# Prevention of mother-to-child transmission of hepatitis B virus: protocol for a one-arm, open-label intervention study to estimate the optimal timing of tenofovir in pregnancy

Marieke Bierhoff [ORCID],[1,2] Kenrad E Nelson,[3] Nan Guo,[4] Yuanxi Jia,[3] Chaisiri Angkurawaranon,[5] Podjanee Jittamala,[6,7] Verena Carrara,[1,8] Wanitda Watthanaworawit,[9] Clare Ling,[8,9] Fuanglada Tongprasert,[10] Michele van Vugt,[2] Marcus Rijken,[11] Francois Nosten [ORCID],[1,8] Rose McGready,[1,8] Stephan Ehrhardt,[3] Chloe Lynne Thio[12]

SE and CLT contributed equally.

For numbered affiliations see end of article.

**Correspondence to**
Dr Stephan Ehrhardt;
sehrhar6@jhu.edu

## ABSTRACT

**Introduction** Hepatitis B virus (HBV) remains a public health threat and the main route of transmission is from mother to child (MTCT). Tenofovir disoproxil fumarate (TDF) treatment can reduce MTCT of HBV although the optimal timing to attain undetectable HBV DNA concentrations at delivery is unknown. This protocol describes the procedures following early initiation of maternal TDF prior to 20 weeks gestation to determine efficacy, safety and feasibility of this approach in a limited-resource setting.

**Methods and analyses** One hundred and seventy pregnant women from the Thailand–Myanmar border between 12 and <20 weeks gestational age will be enrolled into a one-arm, open-label, TDF treatment study with cessation of TDF 1 month after delivery. Sampling occurs monthly prenatal, at birth and at 1, 2, 4 and 6 months post partum. Measurement of tenofovir concentrations in maternal and cord plasma is anticipated in 10–15 women who have detectable HBV DNA at delivery and matched to 20–30 women with no detectable HBV DNA. Infant HBsAg status will be determined at 2 months of age and HBV DNA confirmed in HBsAg positive cases. Adverse events including risk of flare and adherence, based on pill count and questionnaire, will be monitored. Infants will receive HBV vaccinations at birth, 2, 4 and 6 months and hepatitis B immunoglobulin at birth if the mother is hepatitis B e antigen positive. Infant growth and neurodevelopment at 6 months will be compared with established local norms.

**Ethics and dissemination** This study has ethical approval by the Ethics Committee of the Faculty of Tropical Medicine, Mahidol University (FTM ECF-019-06), Johns Hopkins University (IRB no: 00007432), Chiang Mai University (FAM-2559-04227), Oxford Tropical Research Ethics Committee (OxTREC Reference: 49-16) and by the local Tak Community Advisory Board (TCAB-02/REV/2016). The article will be published as an open-access publication.

**Trial registration number** NCT02995005, Pre-results.

## Strengths and limitations of this study

► This prospective cohort study will inform the optimal time to initiate tenofovir disoproxil fumarate (TDF) treatment in pregnant women to reduce hepatitis B virus (HBV) DNA to undetectable concentrations (defined as <100 IU/mL) prior to birth, which is a strength since this is currently unknown.

► Strategies for prevention of mother-to-child transmission (MTCT) of HIV are not immediately transferable to HBV so another strength is that this study investigates the feasibility of implementing the recommended treatment for HBV in pregnant women in a limited-resource setting.

► It is important to identify challenges to adherence and safety in a context of low literacy where two-thirds of pregnant women have limited education to ensure that potential implementation of this strategy is feasible

► This study investigates TDF usage during pregnancy for prevention of MTCT as a potential alternative to hepatitis B immunoglobulin (HBIG), which remains too expensive and difficult to obtain in most limited-resource settings, and will inform future studies that forgo HBIG.

► This study investigates TDF in addition to HBIG and vaccination in infants born to hepatitis B e antigen-positive mothers as this is the most comprehensive way to prevent MTCT, but such a design does not provide information on the proportion of MTCT in women with an undetectable HBV DNA without HBIG.

## INTRODUCTION
### Background and rationale

In 2016, it was estimated that only 1 in 10 of the 257 million people living with hepatitis B virus (HBV) were aware of their infection and

of those, 1 in 7 were on treatment, which is less than 2% of the HBV infections globally.[1] The current sustainable development goals aim to reduce the prevalence of HBV by 90% by 2030.[2] An essential route to reach this goal is prevention of mother-to-child transmission (MTCT) of HBV, the most common source of infection. Pregnancy provides a window of opportunity in the life of an adult to advocate for a screening intervention, raising awareness of HBV infection, potentially resulting in positive impacts on prevalence. Unfortunately, both screening and prevention efforts are impeded by lack of access to universal healthcare and essential health services, a problem relevant to half of the world's population.[3]

Maternal HBV infection is an important communicable disease on the Myanmar–Thailand border[4] and in other parts of Asia, the western Pacific and African Region, where more than 6% of adults are infected.[1] In antenatal clinics at the Shoklo Malaria Research Unit (SMRU) clinics, the hepatitis B surface antigen (HBsAg) prevalence is 6.4%–8.3% with a hepatitis B e antigen (HBeAg) prevalence of 32.7% in those positive for HBsAg.[4 5] Mothers who are HBeAg positive are at the highest risk of transmitting HBV to their infants, which is the basis for specifically targeting this group for neonatal hepatitis B immunoglobulins (HBIG) within 72 hours of life. In infants born to HBeAg-positive mothers and provided HBIG and HBV vaccine, prevention of MTCT fails in an estimated 8%–32% of cases.[6] Overall, there are two reasons to find alternatives to HBIG: (1) failure to prevent MTCT in infants born in HBeAg-positive women, and (2) for the 50% of the world's population without universal healthcare who lack access to HBIG as it remains a specialised product that is costly, requires a cold chain and has a restrictive prescription window to be effective.[7]

The usage of antivirals for HIV in limited-resource settings has shown remarkable scale up and success and is a vital component of antenatal care (ANC) programmes.[8] Antivirals that are active against HBV like tenofovir disoproxil fumarate (TDF) may reduce the risk of MTCT by reducing the HBV DNA across gestation to a minimum at the time of delivery when MTCT typically occurs and might be a more feasible option than HBIG in a limited-resource setting.[9] The optimal gestational age to commence TDF is critical to know to ensure HBV DNA is as low as possible prior to birth.[10] The precise threshold of HBV DNA, in the absence of HBIG that prevents MTCT, is also still unknown. Initially, an HBV DNA of ≥6 log10 IU/ml was associated with a higher risk of MTCT, suggesting that women in this category were the most important to target for antiviral therapy.[11–13] Although a maternal HBV DNA <6 log10 IU/ml is thought to be lower risk for MTCT, the studies reporting this provided the infants with HBIG and vaccination. Although provision of TDF 300 mg daily in the third trimester of pregnancy significantly reduced perinatal transmission (OR 0.10; 95% CI 0.01 to 0.77; p=0.03),[6 14–16] a mean of 8.6±0.5 weeks of TDF prior to delivery resulted in a mean HBV DNA at delivery of 4 log 10 IU/mL with 32% of mothers

having HBV DNA >5.3 log 10 (or 200 000 IU/ml).[6] This amount of HBV DNA is too high to forego HBIG, and if provided, may still not be sufficient to prevent infection. In the same study, HBV DNA declined another log (median 3.52 log10 IU/ml) in the 4 weeks post partum that TDF was provided. Gao *et al* showed that starting TDF treatment at an estimated gestational age of 24–27 weeks may achieve better suppression compared with starting in third trimester.[16]

Existing data from trials involving non-pregnant patients are not informative due to differences in age, stage of HBV, frequency of HBV DNA testing and immune response and pharmacokinetics of TDF in pregnant women compared with non-pregnant patients.[17–20] Pharmacokinetic studies have established that the area under the curve and maximum drug concentration ($C_{max}$) of TDF are decreased in the second and third trimester of pregnancy, and clearance is increased.[18 21–24] In a systematic review of TDF pharmacokinetics in pregnancy, none of the manuscripts reported reaching the TDF $C_{max}$ of 300 ng/mL the efficacy threshold determined in non-pregnant HBV infected women.[23] These data suggest that TDF dosage and treatment initiation that are accurate for HIV and non-pregnant HBV-infected patients may not be accurate for pregnant HBV-infected women.

While some of the safety issues have been resolved with the use of TDF for prevention of MTCT of HIV, gaps remain regarding HBV.[25 26] The main barriers to implementation of TDF are a (1) lack of knowledge by the women of hepatitis B, of their infection status, and of prevention strategies,[27] (2) poor adherence to the prophylactic drug regimen,[9] and (3) potential for hepatitis flares after postpartum drug discontinuation.[28] All of these need to be evaluated and addressed in limited-resource settings where HBV is highly endemic.[29]

The aim of this prospective interventional study is to determine the appropriate time in pregnancy to initiate TDF treatment of HBV in order to reach HBV DNA <100 IU/mL at delivery. Knowing this will allow future studies where TDF can be initiated at the appropriate time so that HBIG may not be needed to prevent MTCT of HBV. The viral kinetics of HBV DNA in pregnant women receiving daily oral TDF commenced before 20 weeks gestation will be measured monthly. Infants will receive HBIG and birth-dose monovalent HBV vaccine. MTCT will be assessed at 2 months post partum. While the study population is pregnant women attending antenatal and birth services on the Myanmar–Thailand border, the study has worldwide application to women who are pregnant and HBV positive. Moreover, we will study safety, feasibility and potential barriers to implementation of TDF in routine ANC in a limited-resource setting.

## Objectives

The primary objective is to estimate the time to HBV DNA suppression <2 log10 IU/mL (or <100 IU/mL) in HBV DNA positive women who start TDF late in the first or early in the second trimester.

Secondary objectives include: estimation of the proportion of women with HBV DNA <2 log10 IU/mL (or <100 IU/mL) at delivery; the concentration of TDF in maternal and cord blood at birth in women who have detectable HBV DNA and a subset of women with undetectable HBV DNA; to monitor the safety of TDF on the Myanmar–Thailand border in pregnant women with low health literacy rates[30] and address potential barriers to implementing TDF in early pregnancy to prevent MTCT of HBV; to determine the rate of hepatic flares post partum after cessation of TDF; to estimate the proportion of cases of vertical transmission in infants of 2 months of age; to assess fetal growth by monthly ultrasound, and infant growth at 1, 2, 4 and 6 months and neurodevelopment at 6 months.

## Study design

This is a prospective, one-arm, open-label, intervention study. Women who enrol in the study will be provided with a daily 300 mg dose of oral TDF starting at enrolment until 1 month after delivery. Follow-up will continue at 1, 2, 4 and 6 months post partum. A small subgroup of women are expected to require ongoing treatment for HBV flares and they will be followed for a further 6–9 months and provided drug therapy if required.

## METHODS AND ANALYSIS
### Setting

The study will be set in the clinics of SMRU on the border of Thailand and Myanmar (figure 1). Since the 1990s, the migrant influx from Myanmar to Thailand has increased for economic and political reasons, and to date there are approximately 3.5 million migrants in Thailand.[31] Over 25% of the migrant workers have no work permits and no legal status in Thailand.[32] Migrants are highly mobile, and the majority do not have access to basic healthcare which, combined with scarce services in Myanmar, put them at increased risk when seeking health services. In

rural areas, labour migrants may be paid daily wages below Thailand's legal minimum wage reducing health-seeking behaviours which incur cost.[33]

SMRU has provided humanitarian healthcare for marginalised populations for 30 years. One of the original objectives was to reduce malaria-related maternal mortality from *Plasmodium falciparum* with screening embedded into routine maternal and child healthcare. The study will invite participants from three clinic sites (Mawker Thai, Wang Pha and Maela) serving marginalised populations (figure 1). The long experience of SMRU working with the population has resulted in a high level of trust in the services. The maternal and child health services are in rural areas, can be provided in Karen and Burmese language and is without costs.[5 34 35]

At the time of study protocol submission (August 2017), antiviral treatment of HBV in pregnancy was not provided by the government in Myanmar or in Thailand, although it could be obtained by out-of-pocket expenditure beyond the means of this marginalised border population.

## Study recruitment and enrolment

Pregnant women aged 16–49 years with a singleton viable pregnancy, an HBV infection, and an estimated gestational age of 12 to <20 weeks determined by ultrasound will be counselled about the disease and invited to participate in the study. Part of this information session will include counselling on HBV in general, the need for adherence to the study drug and benefits and risks of the treatment. If a woman is interested, a blood sample will be collected to determine if she meets the criteria for enrolment, that is, HBeAg positive or HBV DNA >1.9 log 10 (>85 IU/mL). Women will be informed of the results at the next visit and be enrolled and started on TDF if all inclusion criteria are met (table 1).

SMRU has conducted research for over two decades and the trust in antenatal services is high resulting in high follow-up rates in ANC as well as in previous studies.[36] If a woman misses an appointment, she will be contacted using her phone number if applicable or the healthcare workers will reach out to her community or family to find her.

## Follow-up
### Medical and obstetric data

Vital signs (blood pressure, tympanic temperature, heart rate, respiratory rate, fetal heart rate); weight, height, age, symphysis fundal height, fetal gestational age; medical history review (including medications, vitamins and allergies), ethnicity, details of any past pregnancies, complications, place of delivery and outcome will be recorded; and physical examination completed (figure 2 and table 2).

### Laboratory tests

A maternal whole blood sample obtained by venipuncture will be collected at study entry, monthly until delivery, at birth and at two postpartum visits (figure 2). Maternal HBV DNA, alanine aminotransferase (ALT) and creatinine will be measured at all scheduled visits for the

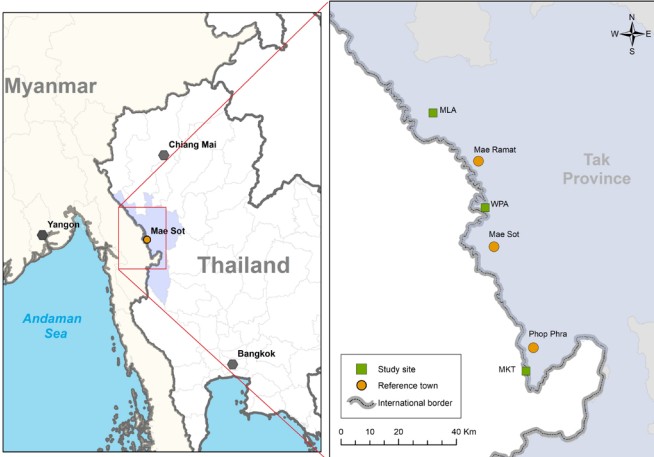

**Figure 1** Map of the central office of Shoklo Malaria Research Unit in Mae Sot, and the study sites Wang Pha (WPA), Mawker Thai (MKT) and Maela (MLA) on the Myanmar–Thailand border where the study will be conducted.

**Table 1** Inclusion and exclusion criteria

| Inclusion criteria | Exclusion criteria |
| --- | --- |
| Willing and able to give informed consent for participation in the study | Negative qualitative HBV DNA if HBeAg negative |
| Hepatitis B infected (HBsAg confirmed positive and qualitative HBV DNA positive in HBeAg-negative mothers) | HIV positive or on immunosuppressive therapy for other illnesses |
| Female, 16–49 years | Elevated creatinine (>1 mg/dL) |
| EGA 12–20 weeks at start TDF | Abnormal serum phosphate (<2.4 and >4.5 mg/dL) |
| Willing to take TDF daily during pregnancy | History of kidney disease |
| Plans to deliver at SMRU | History of pregnancy complications |

EGA, estimated gestational age; HBeAg, hepatitis B e antigen; HBsAg, hepatitis B surface antigen; SMRU, Shoklo Malaria Research Unit; TDF, tenofovir disoproxil fumarate.

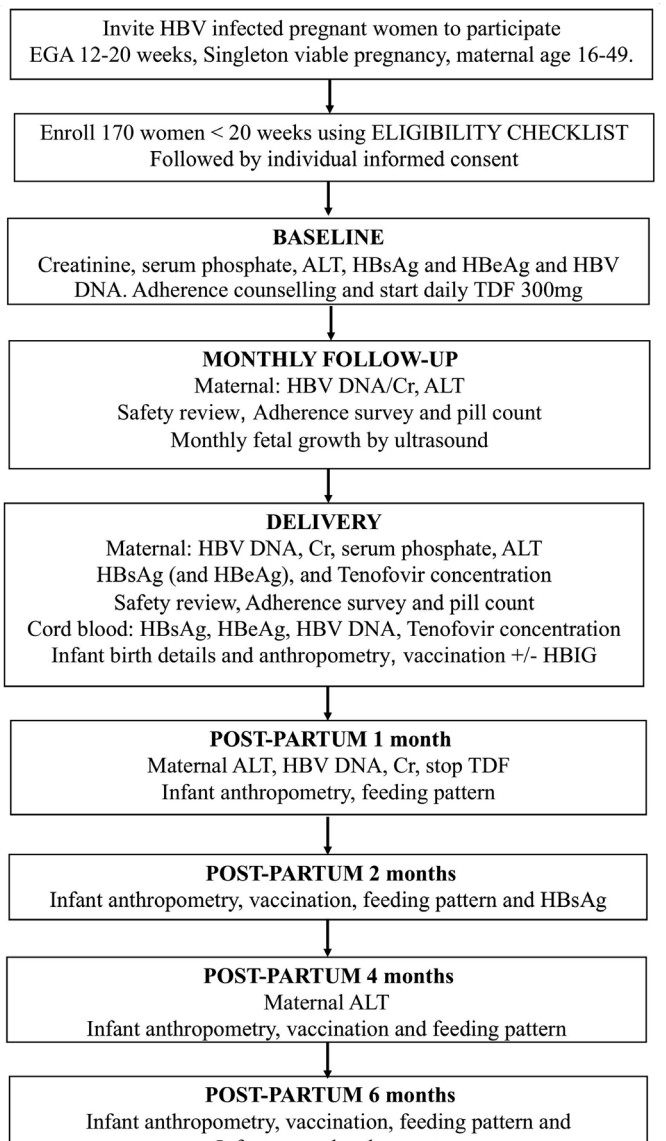

**Figure 2** Schematic reproduction of the study design. ALT, alanine aminotransferase; Cr, creatinine; EGA, estimated gestational age; HBeAg, hepatitis B envelope antigen; HBIG, hepatitis B immunoglobulins; HBsAg, hepatitis B surface antigen; HBV, hepatitis B virus; TDF, tenofovir disoproxil fumarate.

mother. Serum phosphate will be measured at baseline and at birth.

## Monthly visits during pregnancy

From the time TDF commences, the pregnant women will return for a monthly study visit. Obstetric follow-up at each monthly visit will include vital signs, weight, symphysis fundal height measurement and fetal heart beat and concomitant medications will be recorded into the case report form (CRF). Fetal growth will be measured monthly by ultrasound (head circumference, abdominal circumference and femur length).

Adverse events (AE) will be reviewed, and reported as AE or serious AE according to the problem. Compliance will be monitored by pill counting and a questionnaire on adherence.

## Delivery

Pregnancy complications and mode of delivery will be recorded in the CRF. Anthropometric measurements of the infant will be done in the first hour of life and will include birth weight, length, head and arm circumference, by trained staff and according to standard operating procedures.

Vaccination: HBIG (if mother is HBeAg positive) and standard HBV vaccine (at birth and 2, 4 and 6 months of age) will be provided to the infants according to the Expanded Program on Immunization.[37–39]

At birth, a 12 mL sample of cord blood, 6 mL into EDTA and 6 mL into a plain tube will be collected for analysis of HBV DNA and tenofovir concentration and HBsAg reflecting the HBV status of the infant. Note this cord blood sample might not be collected if the woman is referred for caesarean section into the public hospital system.

A maternal serum sample will be used to measure ALT, creatinine, phosphorus, HBsAg (HBeAg only if the woman was HBeAg positive at baseline) and tenofovir concentration.

## Postpartum follow-up

TDF will be continued until 1 month post partum to decrease the risk of flares as the maternal immune system returns to baseline. All women will continue the

**Table 2** Overview of measurements and interventions per visit

| Procedures | Start | Monthly visits | Delivery | 1 month PP | 2 months PP | 4 months PP | 6 months PP |
|---|---|---|---|---|---|---|---|
| | | On TDF | On TDF | Stop TDF | | End mother | End infant |
| Eligibility checklist | X | | | | | | |
| Informed consent | X | | | | | | |
| Ultrasound | X | X | X | | | | |
| Demographics | X | | | | | | |
| Medical history | X | | | | | | |
| Physical examination | X | X | X | X | | X | |
| **Laboratory tests** | | | | | | | |
| HBsAg/HBeAg | X | | | | | | |
| HBV DNA | X | X | X | X | | | |
| Creatinine | X | X | X | X | | | |
| Serum phosphate | X | | X | | | | |
| ALT | X | X | X | X | | X* | |
| TDF concentration | | | X† | | | | |
| Cord HBsAg/HBV DNA | | | X† | | | | |
| Infant HBsAg | | | | | X‡ | | |
| Pill count | | X | X | X | | | |
| Adverse event assessments | | X | X | X | | | |
| Infant anthropometry | | | X | X | X | X | X |
| Infant neurodevelopment | | | | | | | X |
| Adherence survey | | X | X | X | | | |
| Infant HB vaccination | | | X§ | | X | X | X |

*Follow-up will continue if suspected or confirmed flare.
†Cord blood sample.
‡Venepuncture sample blood and only for hepatitis B surface antigen (HBsAg).
§If mother hepatitis B e antigen (HBeAg) positive also hepatitis B immunoglobulins (HBIG).
ALT, alanine aminotransferase; HBV, hepatitis B virus; PP, post partum; TDF, tenofovir disoproxil fumarate.

study until the 4 month postpartum visit. At the 1 and 4 months postpartum visits, ALT will be measured. If ALT concentrations are increased, but there is not a true flare, follow-up will continue and repeat ALT testing will be done at 6 months. If the flare worsens or does not resolve, they will be placed on anti-HBV therapy and be reviewed by a physician.

All infants will be followed at 1, 2, 4 and 6 months. The infant will be screened for HBV at 2 months. At each visit, infant anthropometry will be measured (body weight, length, head circumference) and feeding recorded. At 6 months, neurodevelopment will be assessed. At months 2, 4 and 6, the infant will receive follow-up HBV vaccinations.

### Laboratory-based stopping criteria

ALT: monthly lab results will be compared with baseline. Grade 3 or higher elevations will be an indication to stop the drug; grade 3 or higher is >5× upper limit of normal (ULN) if baseline levels were below the ULN; or if baseline levels were >ULN, then grade 3 or higher is >3.5× the baseline value. The ULN of ALT will be defined as (0–31 U/L).[40]

Creatinine: a creatinine value of more than 1.0 mg/dL will be an indication to stop the drug regardless of the baseline value.[40]

### Sample size

We calculated the sample size required to estimate the proportion of pregnant women expected to be undetectable at delivery with precision.

$$n = \frac{(1.96)^2 p(1-p)}{d^2}$$

There is no prior knowledge to predict the proportion of pregnant women expected to be undetectable at delivery, so we used the most conservative estimate. When this proportion is 50%, the required sample size is the largest. Based on a 95% CI, a margin of error d=8% and a conservative estimation of proportion p=0.5, a sample

size of n=150 is required. We aim for n=170 to allow for attrition.

## Data analyses
### HBV viral kinetics
The Kaplan-Meier method will be used to estimate the median time from initiation of TDF to an HBV DNA <100 IU/mL including all women and then, based on the distribution of HBV DNA, stratified by high versus low HBV DNA groups. We expect those with low HBV DNA at enrolment to have <100 IU/mL at the time of delivery, but it is important to include them in this study to calculate the trajectory of decline and determine how long they need to be treated to reach HBV DNA <100 IU/mL. A generalised estimating equation model will be used to assess whether baseline HBV DNA and adherence to TDF are associated with the logarithmically transformed HBV DNA over time. A proportional hazard model will be constructed to assess the HR of reaching undetectable HBV DNA adjusting for the baseline HBV DNA concentration and adherence to TDF (# pills actually taken/# pills that should have been taken). For missing HBV DNA data, depending on the reasons for being missing, imputations using multiple imputations by chained equations will be used. Sensitivity analyses will then be performed comparing analyses using complete and imputed data sets. The proportion of women with undetectable HBV DNA at delivery will be reported with 95% CI in all women and by high versus low HBV DNA groups. Median and IQR of HBV DNA at baseline and each visit will be reported. Then, HBV DNA will be logarithmically transformed and the median HBV DNA at baseline and at delivery will be compared using paired t-tests. Additional analyses may be conducted to further explore the data.

### Adherence
The main analysis to determine adherence will be calculating per cent adherence using pill counting data (# pills actually taken/# pills that should have been taken). As a confirmation of adherence, we will also measure TDF concentration in all women with detectable HBV DNA at delivery and in a matched subset of women who become undetectable for comparison (n=2× the number of women who do not become undetectable). This is anticipated to include 10–15 women who are HBV DNA detectable at delivery and matched to 20–30 women who are undetectable. This analytical subsample will be matched on age and initial DNA concentration or, if a matched pair analysis is not feasible because no 'matching partners' exist, we will correct for these factors using multivariate regression methods.

The 'Adherence Starts with Knowledge' questionnaire ASK-12 is a tool to determine adherence and is derived from the ASK-20 questionnaire.[41] The questionnaire consists of 12 items related to drug adherence in three scales 'forgetfulness', 'treatment beliefs' and 'behavior'. Although previous studies have demonstrated the usefulness of ASK-12 for patients, this tool has not

been validated in this study population.[42 43] Local pilot testing has resulted in a culturally appropriate version of this questionnaire and personnel will be trained to use it. We will use pill counts (gold standard) as well as TDF blood concentration (confirmatory) to validate this tool. The results will be analysed by first comparing differences among the groups, one-way analysis of variance and Bonferroni's multiple comparison tests.

### Hepatic flare
Although TDF is thought to be safe and acceptable in pregnancy, after discontinuation it might lead to hepatitis flares. There are conflicting data about the proportion and severity of flares after discontinuing antiviral therapy.[44–46] There are no data in pregnant women regarding flares with longer duration of treatment and lower HBV DNA at delivery, nor is there information about predisposing characteristics to provide recommendations either before, or even after, flares start. We will determine the proportion of woman who have a moderate (ALT >2.5× ULN) or severe (ALT >5× ULN) hepatitis flare after discontinuation of TDF.[47] We will estimate the impact of potential risk factors such as age, number of prior pregnancies and so on for moderate or severe flares by including treatment duration and HBV DNA concentration at study start and at delivery into multivariable regression models.

### Secondary analyses
We will screen for HBV infection by screening for HBsAg in infants at 2 months of age. If HBsAg is positive, the presence of HBV DNA from the same blood sample will be confirmed. If MTCT of HBV exists, the time of TDF initiation, maternal baseline HBV DNA and TDF adherence will be compared between mothers with MTCT and those without using t-test and $\chi^2$ test/Fisher's exact test. Infant anthropometry will be reported at 1, 2, 4 and 6 months. We will compare the infant anthropometry to the same age infant in general population. Additional analyses such as correlation between TDF use time and infant anthropometry will be conducted based on the data we observe.

## ETHICS AND DISSEMINATION
### Ethical considerations
SMRU has worked with pregnant women for three decades and provided research results that have had significant local and international impact.[30 48] Maternal and infant mortality have decreased significantly over the years SMRU has worked with the population. Women come to deliver with SMRU by choice. These marginalised populations and their ability to negotiate the health landscape may be limited by education level but this is recognised by local health staff who are from the same community and form part of the research team.[30 48] All communication is conducted in the preferred language of the patient and additional efforts are made to assist non-literate women. We ensure that participation is voluntary and that it is

clearly understood that participation in the study can be ceased at any time without effect to the care provided. Extra precautions to inform participants about discontinuation while under TDF are also part of the information sheets.

The Tak Community Advisory Board (T-CAB) reviewed the project for relevance, advice and acceptability in a meeting in relation to this study prior to submission of the protocol.[49] Pertinent responses from the T-CAB in relation to this study were positive but also included a request to include adolescent pregnant women. Several T-CAB members discussed further the issue of consent and assent in relation to SMRU staff, pregnant women, study participants and guardians to further inform the study team on how this issue is best managed with the marginalised, mobile border population.

This study has been approved by the Ethics Committee of the Faculty of Tropical Medicine, Mahidol University (FTM ECF-019-06), Johns Hopkins University (IRB no: 00007432), Chiang Mai University (FAM-2559-04227), Oxford Tropical Research Ethics Committee (OxTREC Reference: 49-16) and by the local T-CAB (TCAB-02/REV/2016).

**Author affiliations**
[1]Department of Maternal and Child health, Shoklo Malaria Research Unit, Mahidol-Oxford Tropical Medicine Research Unit, Faculty of Tropical Medicine, Mahidol University, Mae Sot, Thailand
[2]Amsterdam UMC, Internal Medicine, Department of Infectious Diseases, Centre of Tropical Medicine and Travel Medicine, location AMC, University of Amsterdam, Amsterdam, The Netherlands
[3]Department of Epidemiology, Johns Hopkins University, Baltimore, Maryland, USA
[4]Department of Anesthesiology, Perioperative and Pain Medicine, Stanford University, Stanford, California, USA
[5]Department of Family Medicine, Chiang Mai University, Suthep, Chiang Mai, Thailand
[6]Department of Maternal and Child health, Mahidol-Oxford Tropical Medicine Research Unit, Faculty of Tropical Medicine, Mahidol University, Bangkok, Thailand
[7]Department of Tropical Hygiene, Mahidol University Faculty of Medicine Ramathibodi Hospital, Bangkok, Thailand
[8]Centre for Tropical Medicine and Global Health, Oxford University, Oxford, UK
[9]Department of Microbiology, Shoklo Malaria Research Unit, Mahidol-Oxford Tropical Medicine Research Unit, Faculty of Tropical Medicine, Mahidol University, Mae Sot, Thailand
[10]Department of Obstetrics and Gynecology, Chiang Mai University, Suthep, Chiang Mai, Thailand
[11]Department of Obstetrics and Gynecology, Utrecht University, Utrecht, The Netherlands
[12]Department of Medicine, Johns Hopkins University School of Medicine, Baltimore, Maryland, USA

**Acknowledgements** The authors would kindly like to acknowledge all contributions of SMRU staff and doctors for their feedback in finalising the research proposal. Also, the data management team for contributing to the Case Report Form development and our quality control team, Moo Kho Paw and Rattanaporn Raksapraidee, and clinical trials support group, for their efforts. Myo Chit Min helped to design the map.

**Contributors** KEN, SE, CLT and RM designed the protocol. SE, KEN and CLT obtained funding. RM, CLT, SE, FT, CA and MB obtained the ethical approvals. RM and MB set up the clinical work. WW and CLL prepared laboratory testing. NG and YJ supported statistical considerations. PJ, VC, FT, MvV, MR and FN supported the design and clinical implementation of the study. All authors read and contributed to the present manuscript. All authors read and approved the final manuscript.

**Funding** The work is supported by the following awards: Thrasher Research Fund (grant number: TRF13096) to Johns Hopkins University; the Wellcome-Trust Major Overseas Program in Southeast Asia (grant number: 106698/Z/14/Z) to Mahidol University Oxford Tropical Medicine Research Programme which directly supports PJ, CLL, FN and RM from Shoklo Malaria Research Unit; and Chiang Mai University Thailand supporting MB, FT and CA.

**Map disclaimer** The depiction of boundaries on this map does not imply the expression of any opinion whatsoever on the part of BMJ (or any member of its group) concerning the legal status of any country, territory, jurisdiction or area or of its authorities. This map is provided without any warranty of any kind, either express or implied.

**Competing interests** None declared.

**Patient and public involvement** Patients and/or the public were involved in the design, or conduct, or reporting, or dissemination plans of this research. Refer to the Methods section for further details.

**Patient consent for publication** Not required.

**Provenance and peer review** Not commissioned; externally peer reviewed.

**Data availability statement** Data sharing not applicable as no data sets generated and/or analysed for this study. After analyses of the data, the data will be made available.

**ORCID iDs**
Marieke Bierhoff http://orcid.org/0000-0003-2585-1066
Francois Nosten http://orcid.org/0000-0002-7951-0745

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
