## [Reviewer comments · BMJ Open]

ARTICLE DETAILS

TITLE (PROVISIONAL)	Prevention of mother-to-child transmission of hepatitis B virus: protocol for a one arm, open label intervention study to estimate the optimal timing of tenofovir in pregnancy
AUTHORS	Bierhoff, Marieke; Nelson, Kenrad; Guo, Nan; Jia, Yuanxi; Angkurawaranon, Chaisiri; Jittamala, Podjane; Carrara, Verena; Watthanaworawit, Wanitda; Ling, Clare; Tongprasert, Fuanglada; van Vugt, Michele; Rijken, Marcus; Nosten, Francois; McGready, Rose; Ehrhardt, Stephan; Thio, Chloe

VERSION 1 - REVIEW

REVIEWER	Manjeetha Jaggernath Match Research Unit Wits Health Consortium South Africa
REVIEW RETURNED	12-Apr-2020

GENERAL COMMENTS	1. What is the GRF cut off for discontinuing PrEP in pregnant women?2. The sentence in Line 43/44 starting with "Although reports..." doesn't seem to make sense? is there a word missing?3. Line 21: the abbreviation is listed as MCTC - it should be MTCT.
---

REVIEWER	Jillian Henderson Kaiser Permanente, Center for Health Research, Portland, Oregon, USA
REVIEW RETURNED	23-Apr-2020

GENERAL COMMENTS	This is a protocol for a single-arm cohort study investigating the outcomes associated with tenofovir treatment initiated in the late first or early third trimester of pregnancy for reducing the risk of mother-to-child transmission. The study is clearly described and well-designed to address the research objective. The protocol would be strengthened with some description to support the authors' assumption of a very high followup rate for enrolled study participants. Have other studies conducted in this setting attained such high followup? Is there any reason to doubt that these expectations could be overly optimistic, and if so what plans are in place to address any problems with power that would arise in the event of loss to followup higher than planned for?
---

	The protocol is otherwise clear and complete, and if the study is completed to protocol will be an important contribution to the science on this topic.
--	---

REVIEWER	Xuesong Gao Beijing Ditan Hospital, Capital Medical University, Beijing, China
REVIEW RETURNED	19-May-2020

GENERAL COMMENTS	The aim of this study is to determine the optimal time to initiate TDF treatment of HBV in pregnancy in order to prevent MTCT. It is helpful to improve the blocking rate of MTCT. The following point need to be considered: The inclusion criteria (HBeAg positive or HBV DNA >85 IU/mL) are too broad. Women with low HBV loads have lower risk of MTCT and they do not have to initiate TDF in the first or early in the second trimester. The mothers with lower viral load are supposed to commence antiviral treatment lately. The mothers with lower viral loads start antiviral treatment from 12 to 20 weeks, which increases the unnecessary TDF exposure time in pregnancy. Thus, the risk of adverse effect and the cost of treatment will be increased. In addition, the increase of cost may lead to the decrease of the number of mothers with HBV who receive treatment in the low-income area. At the same time, prolonging the course of treatment may reduce the adherence to the treatment for the patients with lower education level. The above reasons are not conducive to promoting the research. I would recommend that the authors enroll mothers with high HBV loads. The mothers should be divided into different groups according viral loads.
---

VERSION 1 – AUTHOR RESPONSE

Reviewer: 1

Reviewer Name: Manjeetha Jaggernath

Institution and Country: Match Research Unit, Wits Health Consortium, South Africa

Please state any competing interests or state 'None declared': None declared

1. What is the GRF cut off for discontinuing PrEP in pregnant women?

On page 11 we added the laboratory based stopping criteria: "Laboratory based stopping criteria
ALT: Monthly lab results will be compared to baseline. Grade 3 or higher elevations will be an indication to stop the drug. Grade 3 or higher is >5x upper limit of normal (ULN) if baseline levels were below the ULN; or if baseline levels were >ULN and , then grade 3 or higher is >3.5x the baseline value. The upper limit of normal of ALT will be defined as (0-31 U/L) [40].

Creatinine: A creatinine value of more than 1.0 mg/dL will be an indication to stop the drug regardless of the baseline value [40]."

2. The sentence in Line 43/44 starting with "Although reports..." doesn't seem to make sense? is there a word missing?

Thank you for this comment, this is changed into "Although previous studies have demonstrated the usefulness of ASK-12 for patients, this tool has not been validated in this study population"

3. Line 21: the abbreviation is listed as MCTC - it should be MTCT.

Thank you, this has been changed

Reviewer: 2

Reviewer Name: Jillian Henderson

Institution and Country: Kaiser Permanente, Center for Health Research, Portland, Oregon, USA

Please state any competing interests or state 'None declared': None declared

This is a protocol for a single-arm cohort study investigating the outcomes associated with tenofovir treatment initiated in the late first or early third trimester of pregnancy for reducing the risk of mother-to-child transmission. The study is clearly described and well-designed to address the research objective. The protocol would be strengthened with some description to support the authors' assumption of a very high followup rate for enrolled study participants. Have other studies conducted in this setting attained such high followup?

On page 10 we added: "SMRU has conducted research for over 2 decades and the trust in antenatal services is high resulting in high follow up rates."

Is there any reason to doubt that these expectations could be overly optimistic, and if so what plans are in place to address any problems with power that would arise in the event of loss to followup higher than planned for?

On page 9 we added: "If a woman misses an appointment she will be contacted using her phone number if applicable or the healthcare workers will reach out to her community or family to find her."

The protocol is otherwise clear and complete, and if the study is completed to protocol will be an important contribution to the science on this topic.

Reviewer: 3

Reviewer Name: Xuesong Gao

Institution and Country: Beijing Ditan Hospital, Capital Medical University, Beijing, China

Please state any competing interests or state 'None declared': None

The aim of this study is to determine the optimal time to initiate TDF treatment of HBV in pregnancy in order to prevent MTCT. It is helpful to improve the blocking rate of MTCT.

The following point need to be considered:

The inclusion criteria (HBeAg positive or HBV DNA >85 IU/mL) are too broad. Women with low HBV loads have lower risk of MTCT and they do not have to initiate TDF in the first or early in the second trimester. The mothers with lower viral load are supposed to commence antiviral treatment lately. The mothers with lower viral loads start antiviral treatment from 12 to 20 weeks, which increases the unnecessary TDF exposure time in pregnancy. Thus, the risk of adverse effect and the cost of treatment will be increased. In addition, the increase of cost may lead to the decrease of the number of mothers with HBV who receive treatment in the low-income area. At the same time, prolonging the course of treatment may reduce the adherence to the treatment for the patients with lower education level. The above reasons are not conducive to promoting the research.

I would recommend that the authors enroll mothers with high HBV loads. The mothers should be divided into different groups according viral loads.

Although we appreciate this suggestion, we have now emphasized that the goal of this study is to determine the viral kinetics of HBV DNA on TDF in pregnancy to achieve HBV DNA <100 IU/ml at

delivery. We hypothesize that women with lower HBV DNA levels will require a shorter period to reach undetectable HBV DNA. Exploring these kinetics is critical for designing future studies where TDF is given earlier in pregnancy to be able to forego the use of HBIg, which is difficult to obtain in many resource-limited settings. This aim is different from other studies that have focused on treating women with high viral loads but still continuing to administer HBIg to the infant. In the analyses of the viral kinetics we will be able to draw conclusions about the optimal duration of TDF therapy for different (both high and low) viral loads. We agree that cost will be an important consideration for a larger trial in the future, which is why we are doing this study to determine how long TDF needs to be given to reach HBV DNA <100 IU/ml at delivery.

VERSION 2 – REVIEW

REVIEWER	Xuesong Gao Beijing Ditan Hospital, Capital Medical University, Beijing,China
REVIEW RETURNED	23-Jun-2020

GENERAL COMMENTS	According the previous studies, the mean HBV DNA level was reduced 2-3 Log ₁₀ IU/mL after 8-12 weeks of LdT treatment in pregnant women. TDF has a more potent antiviral effect. Therefore, antiviral treatment from 12-20 week for women with lower HBV DNA levels is not necessary, although TDF' safety in pregnant women has been demonstrated.
--

VERSION 2 – AUTHOR RESPONSE

Reviewer: 3

Reviewer Name: Xuesong Gao

Institution and Country: Beijing Ditan Hospital, Capital Medical University, Beijing,China

Please state any competing interests or state 'None declared': None declared

According the previous studies, the mean HBV DNA level was reduced 2-3 Log₁₀ IU/mL after 8-12 weeks of LdT treatment in pregnant women. TDF has a more potent antiviral effect. Therefore, antiviral treatment from 12-20 week for women with lower HBV DNA levels is not necessary, although TDF' safety in pregnant women has been demonstrated.

Thank you for your comment. We added on page 11 "HBV viral kinetics": "we expect those with low HBV DNA at enrolment to have <100 IU/ml at the time of delivery, but it is important to include them in this study to calculate the trajectory of decline and determine how long they need to be treated to reach HBV DNA <100 IU/ml." This is an important consideration for resource limited settings where HBIG is generally unavailable.